# Inverted encoding of neural responses to audiovisual stimuli reveals super-additive multisensory enhancement

Zak Buhmann[1]*, Amanda K Robinson[2,3], Jason B Mattingley[2,3], Reuben Rideaux[1,2]

[1]School of Psychology, University of Sydney, Sydney, Australia; [2]Queensland Brain Institute, The University of Queensland, Brisbane, Australia; [3]School of Psychology, The University of Queensland, Brisbane, Australia

## eLife Assessment

Despite the well-established facilitatory effects of multisensory integration on behavioural measures, standard neuroimaging approaches have yet to reliably and precisely identify the corresponding neural correlates. In this **valuable** paper, Buhmann et al. leverage EEG decoding methods, moving beyond traditional univariate analyses, to capture these correlates. They present **solid** evidence that this approach can effectively estimate multisensory integration in humans across a broad range of contexts.

**\*For correspondence:**
zbuh0362@uni.sydney.edu.au

**Competing interest:** The authors declare that no competing interests exist.

**Abstract** A central challenge for the brain is how to combine separate sources of information from different sensory modalities to optimally represent objects and events in the external world, such as combining someone's speech and lip movements to better understand them in a noisy environment. At the level of individual neurons, audiovisual stimuli often elicit super-additive interactions, where the neural response is greater than the sum of auditory and visual responses. However, investigations using electroencephalography (EEG) to record brain activity have revealed inconsistent interactions, with studies reporting a mix of super- and sub-additive effects. A possible explanation for this inconsistency is that standard univariate analyses obscure multisensory interactions present in EEG responses by overlooking multivariate changes in activity across the scalp. To address this shortcoming, we investigated EEG responses to audiovisual stimuli using inverted encoding, a population tuning approach that uses multivariate information to characterise feature-specific neural activity. Participants (n=41) completed a spatial localisation task for both unisensory stimuli (auditory clicks, visual flashes) and combined audiovisual stimuli (spatiotemporally congruent clicks and flashes). To assess multivariate changes in EEG activity, we used inverted encoding to recover stimulus location information from event-related potentials (ERPs). Participants localised audiovisual stimuli more accurately than unisensory stimuli alone. For univariate ERP analyses, we found an additive multisensory interaction. By contrast, multivariate analyses revealed a super-additive interaction ~180 ms following stimulus onset, such that the location of audiovisual stimuli was decoded more accurately than that predicted by maximum likelihood estimation. Our results suggest that super-additive integration of audiovisual information is reflected within multivariate patterns of activity rather than univariate evoked responses.

## Introduction

We exist in a complex, dynamically changing sensory environment. Vertebrates, including humans, have evolved sensory organs that transduce relevant sources of physical information, such as light and

changes in air pressure, into patterns of neural activity that support perception (vision and audition) and adaptive behaviour. Such activity patterns are noisy and often ambiguous, due to a combination of external (environmental) and internal (transduction) factors. Critically, information from the different sensory modalities can be highly correlated because it is often elicited by a common external source or event. For example, the sight and sound of a hammer hitting a nail produces a single, unified perceptual experience, as does the sight of a person's lips moving as we hear their voice. To improve the reliability of neural representations, the brain leverages these sensory relationships by combining information in a process referred to as *multisensory integration*. The existence of such processes heightens perception, e.g., by making it easier to understand a person's speech in a noisy setting by looking at their lip movements (*Sumby and Pollack, 1954*).

Multisensory integration of audiovisual cues improves performance across a range of behavioural outcomes, including detection accuracy (*Bolognini et al., 2005*; *Frassinetti et al., 2002*; *Lovelace et al., 2003*), response speed (*Arieh and Marks, 2008*; *Cappe et al., 2009*; *Colonius and Diederich, 2004*; *Rach and Diederich, 2006*; *Senkowski et al., 2011*), and saccade speed and accuracy (*Corneil et al., 2002*; *Van Wanrooij et al., 2009*). Successful integration requires the constituent stimuli to occur at approximately the same place and time (*Leone and McCourt, 2015*). The degree to which behavioural performance is improved follows the principles of maximum likelihood estimation (MLE), wherein sensory information from each modality is weighted and integrated according to its relative reliability (*Alais and Burr, 2004*; *Ernst and Banks, 2002*; although other processing schemes have also been identified; *Rideaux and Welchman, 2018*). As such, behavioural performance that matches MLE predictions is often seen as a benchmark of successful, optimal integration of relevant unisensory cues.

The ubiquity of behavioural enhancements for audiovisual stimuli suggests there are fundamental neural mechanisms that facilitate improved precision. Recordings from single multisensory (audio-visual) neurons within cat superior colliculus have revealed the principle of inverse effectiveness, whereby the increased response to audiovisual stimuli is larger when the constituent unisensory stimuli are weakly stimulating (*Corneil et al., 2002*; *Meredith and Stein, 1983*). Depending on the intensity of the integrated stimuli, the neural response can be either *super-additive*, where the multisensory response is greater than the sum of the unisensory responses, *additive*, equal to the sum of responses, or *sub-additive*, where the combined response is less than the sum of the unisensory responses (see *Stein and Stanford, 2008*). Inverse effectiveness has also been observed in human behavioural experiments, with low-intensity audiovisual stimuli eliciting greater multisensory enhancements in response precision than those of high intensity (*Colonius and Diederich, 2004*; *Corneil et al., 2002*; *Rach and Diederich, 2006*; *Rach et al., 2011*).

Neuroimaging methods, such as electroencephalography (EEG) and functional magnetic resonance imaging (fMRI), have been used to investigate neural population-level audiovisual integration in humans. These studies have typically applied an additive criterion to quantify multisensory integration, wherein successful integration is marked by a non-linear enhancement of audiovisual responses relative to unisensory responses (*Besle et al., 2004*). The super- or sub-additive nature of this enhancement, however, is often inconsistent. In fMRI, neural super-additivity in blood-oxygen-level-dependent (BOLD) responses to audiovisual stimuli has been found in a variety of regions, primarily the superior temporal sulcus (STS; *Calvert et al., 2000*; *Calvert et al., 2001*; *Stevenson et al., 2007*; *Stevenson and James, 2009*; *Werner and Noppeney, 2010*; *Werner and Noppeney, 2011*). However, other studies have failed to replicate audiovisual super-additivity in the STS (*Joassin et al., 2011*; *Porada et al., 2021*; *Ross et al., 2022*; *Venezia et al., 2015*) or have found sub-additive responses (see *Scheliga et al., 2023*, for review). As such, some have argued that BOLD responses are not sensitive enough to adequately characterise super-additive audiovisual interactions within populations of neurons (*Beauchamp, 2005*; *James et al., 2012*; *Laurienti et al., 2005*). In EEG, meanwhile, the evoked response to an audiovisual stimulus typically conforms to a sub-additive principle (*Cappe et al., 2010*; *Fort et al., 2002*; *Giard and Peronnet, 1999*; *Murray et al., 2016*; *Puce et al., 2007*; *Stekelenburg and Vroomen, 2007*; *Teder-Sälejärvi et al., 2002*; *Vroomen and Stekelenburg, 2010*). However, when the principle of inverse effectiveness is considered and relatively weak stimuli are presented together, there has been some evidence for super-additive responses (*Senkowski et al., 2011*).

It is important to consider the differences in how super-additivity is classified between neural and behavioural measures. At the level of single neurons, super-additivity is defined as a non-linear response enhancement, with the multisensory response exceeding the sum of the unisensory responses. In behaviour, meanwhile, it has been observed that the performance improvement from combining two senses is close to what is expected from optimal integration of information across the senses (*Alais and Burr, 2004*; *Stanford and Stein, 2007*). Critically, behavioural enhancement of this kind does not require non-linearity in the neural response, but can arise from a reliability-weighted average of sensory information. In short, behavioural performance that conforms to MLE is not necessarily indicative of neural super-additivity, and the MLE model can be considered a linear baseline for multisensory integration.

While behavioural outcomes for multisensory stimuli can be predicted by MLE, and single neuron responses follow the principles of inverse effectiveness and super-additivity, among others (*Rideaux et al., 2021*), how audiovisual super-additivity manifests within populations of neurons is comparatively unclear given the mixed findings from relevant fMRI and EEG studies. This uncertainty may be due to biophysical limitations of human neuroimaging techniques, but it may also be related to the analytic approaches used to study these recordings. For instance, super-additive responses to audiovisual stimuli in EEG studies are often reported from very small electrode clusters (*Molholm et al., 2002*; *Senkowski et al., 2011*; *Talsma et al., 2007*), suggesting that neural super-additivity in humans may be highly specific. However, information encoded by the brain can be represented as increased activity in some areas, accompanied by decreased activity in others, so simplifying complex neural responses to the average rise and fall of activity in specific sensors may obscure relevant multivariate patterns of activity evoked by a stimulus.

Inverted encoding is a multivariate analytic method that can reveal how sensory information is encoded within the brain by recovering patterns of neural activity associated with different stimulus features. This method has been successfully used in fMRI, EEG, and magnetoencephalography studies to characterise the neural representations of a range of stimulus features, including colour (*Brouwer and Heeger, 2009*), spatial location (*Bednar and Lalor, 2020*; *Robinson et al., 2021*), and orientation (*Brouwer and Heeger, 2011*; *Harrison et al., 2023*; *Kok et al., 2017*). A multivariate approach may capture potential non-linear enhancements associated with audiovisual responses and thus could reveal super-additive interactions that would otherwise be hidden within the brain's

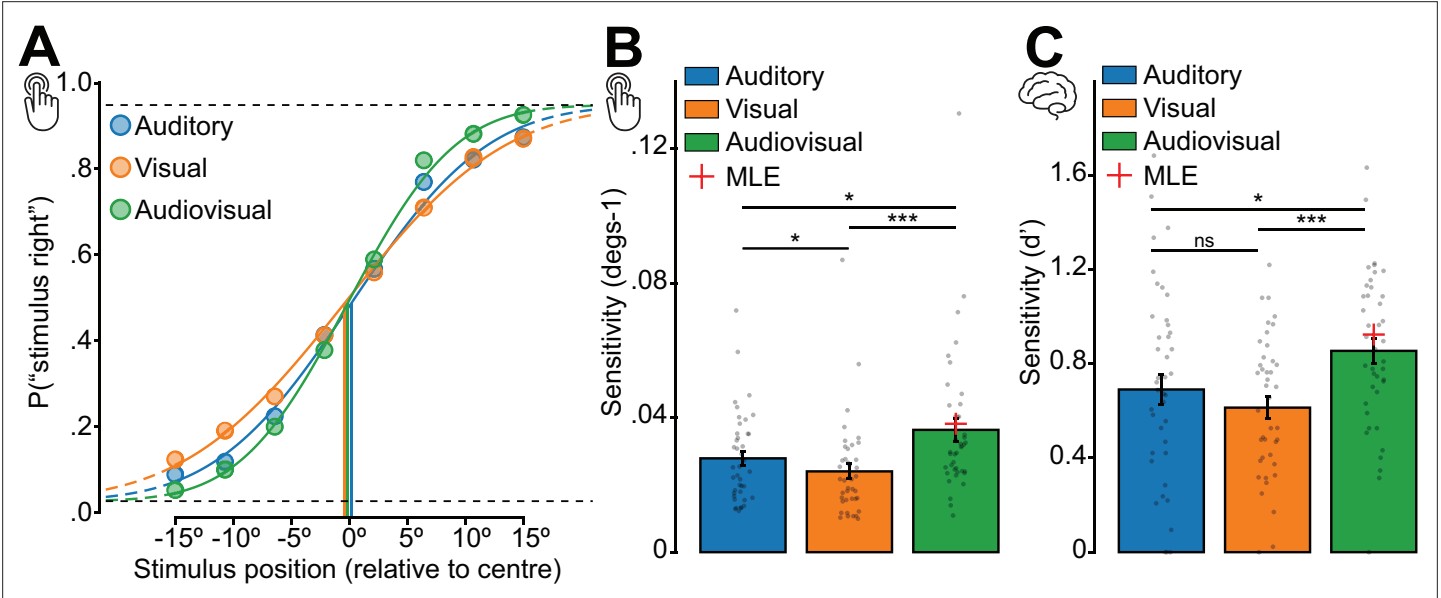

**Figure 1.** Behavioural performance is improved for audiovisual stimuli. (**A**) Average accuracy of responses across participants in the behavioural session at each stimulus location for each stimulus condition, fitted to a psychometric curve. Steeper curves indicate greater sensitivity in identifying stimulus location. (**B**) Average sensitivity across participants in the behavioural task, estimated from psychometric curves, for each stimulus condition. The red cross indicates estimated performance assuming optimal (maximum likelihood estimation [MLE]) integration of unisensory cues. (**C**) Average behavioural sensitivity across participants in the EEG session for each stimulus condition. Error bars indicate ±1 SEM, n = 41.

univariate responses. The sensitivity of inverted encoding analyses to multivariate neural patterns may provide insight into how audiovisual information is processed and integrated at the population level.

In the present study, we investigated neural super-additivity in human audiovisual sensory processing using inverted encoding of EEG responses during a task where participants had to spatially localise visual, auditory, and audiovisual stimuli. In a separate behavioural experiment, we monitored response accuracy to characterise behavioural improvements to audiovisual relative to unisensory stimuli. Although there was no evidence for super-additivity in response to audiovisual stimuli within univariate ERPs, we observed a reliable non-linear enhancement of multivariate decoding performance at ~180 ms following stimulus onset when auditory and visual stimuli were presented concurrently as opposed to alone. These findings suggest that population-level super-additive multisensory neural responses are present within multivariate patterns of activity rather than univariate evoked responses.

## Results

### Behavioural performance

Participants performed well in discriminating stimulus location across all conditions in both the behavioural and EEG sessions (*Figure 1*). For the behavioural session, the psychometric curves for responses as a function of stimulus location showed stereotypical relationships for the auditory, visual, and audiovisual conditions (*Figure 1A*). A quantification of the behavioural sensitivity (i.e. steepness of the curves) revealed significantly higher sensitivity for the audiovisual stimuli (*M*=0.04, SD = 0.02) than for the auditory stimuli alone (*M*=0.03, SD = 0.01; *Z*=–3.09, p=0.002), and than for the visual stimuli alone (*M*=0.02, SD = 0.01; *Z*=–5.28, p=1.288e-7; *Figure 1B*). Sensitivity for auditory stimuli

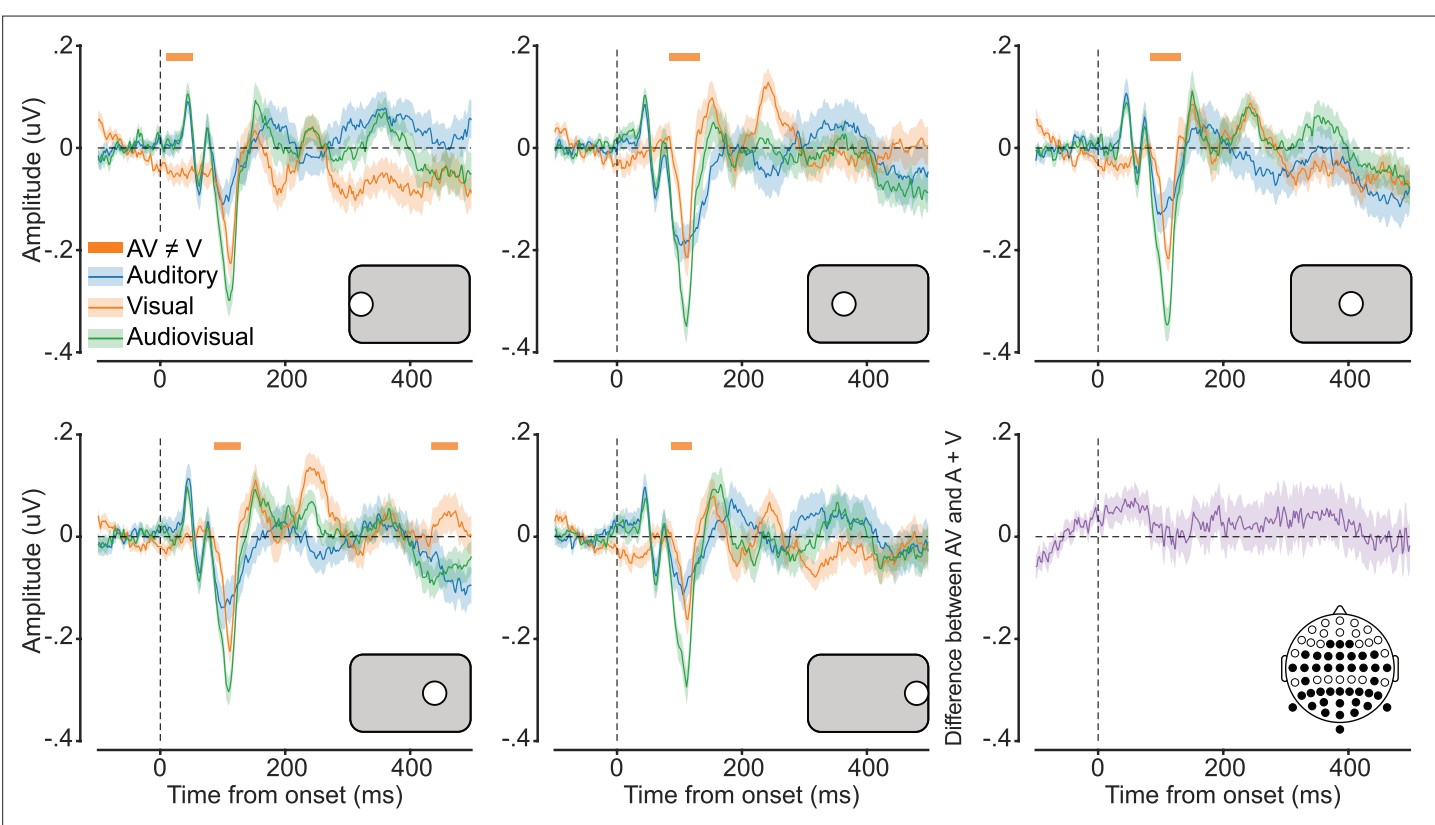

**Figure 2.** Audiovisual event-related potentials (ERPs) follow an additive principle. Average ERP amplitude for each modality condition. Five plots represent the different stimulus locations, as indicated by the grey inset, and the final plot (bottom-right) shows the difference between the summed auditory and visual responses and the audiovisual response. Shaded error bars indicate ±1 SEM, n = 41. Orange horizontal bars indicate cluster corrected periods of significant difference between visual and audiovisual ERP amplitudes.

The online version of this article includes the following figure supplement(s) for figure 2:

**Figure supplement 1.** Correlations reveal a positive relationship between eye position and stimulus position in all conditions.

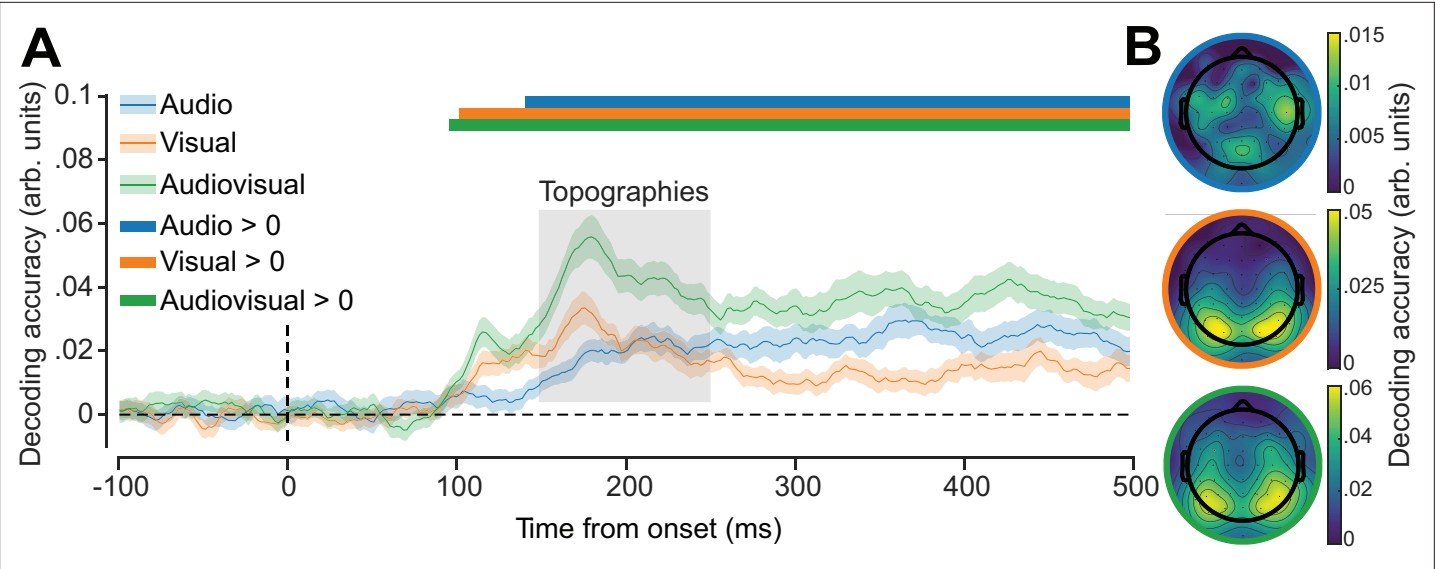

**Figure 3.** Spatiotemporal representation of audiovisual location. (**A**) Accuracy of locations decoded from neural responses for each stimulus condition. Shaded error bars indicate ±1 SEM, n = 41. Coloured horizontal bars indicate cluster corrected periods that showed a significant difference from chance (0). (**B**) Topographic decoding performance in each condition during critical period (grey inset in (**A**)).

was also significantly higher than sensitivity for visual stimuli (*Z*=2.02, p=0.044). To test for successful integration of stimuli in the audiovisual condition, we calculated the predicted MLE sensitivity from the unisensory auditory and visual results. We found no evidence for a significant difference between the predicted and actual audiovisual sensitivity (*Z* = –1.54, p=0.125).

We repeated these analyses for behavioural performance in the EEG session (*Figure 1C*). We found a similar pattern of results to those in the behavioural session; sensitivity for audiovisual stimuli (*M*=0.85, SD = 0.33) was significantly higher than for auditory (*M*=0.69, SD = 0.41; *Z*=–2.27, p=0.023) and visual stimuli alone (*M*=0.61, SD = 0.29; *Z*=–3.52, p=4.345e-4), but not significantly different from the MLE prediction (*Z*=–1.07, p=0.285). However, sensitivity for auditory stimuli was not significantly different from sensitivity to visual stimuli (*Z*=1.12, p=0.262).

## Event-related potentials

We plotted the ERPs for auditory, visual, and audiovisual conditions at each stimulus location from –100 to 500 ms around stimulus presentation (*Figure 2*). For each stimulus location, cluster corrected *t*-tests were conducted to assess significant differences in ERP amplitude between the unisensory (auditory and visual) and audiovisual conditions. While auditory ERPs did not significantly differ from the audiovisual, visual ERPs were significantly lower in amplitude than audiovisual ERPs at all stimulus locations (typically from ~80 to 130 ms following stimulus presentation).

To test whether the enhancement in response amplitude to audiovisual stimuli was super-additive, we compared this response with the sum of the response amplitudes for visual and auditory conditions, averaged over stimulus location. We found no significant difference between the additive and audiovisual ERPs (*Figure 2*, bottom-right). This result suggests that, using univariate analyses, the audiovisual response was additive and did not show any evidence for super- or sub-additivity.

## Inverted encoding results

We next used inverted encoding to calculate the spatial decoding accuracy for auditory, visual, and audiovisual stimuli (*Figure 3A*). For all conditions, we found that spatial location could be reliably decoded from approximately ~100 to 150 ms after stimulus onset. Decoding for all conditions was consistent for most of the epoch, indicating that location information within the neural signal was relatively persistent and robust.

To assess the spatial representation of the neural signal containing location-relevant information, we computed decoding accuracy at each electrode from 150 to 250 ms post-stimulus presentation

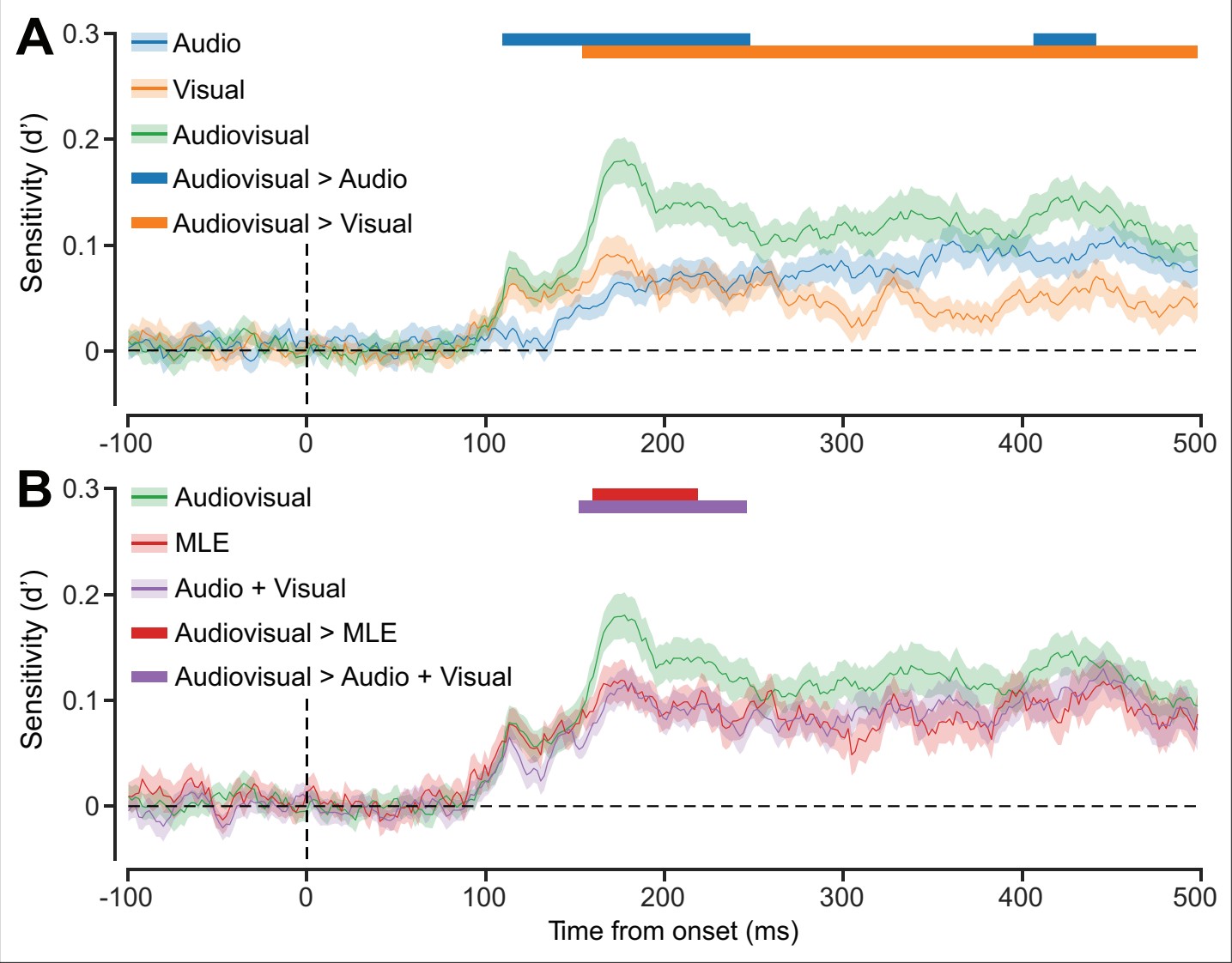

**Figure 4.** Super-additive multisensory interaction in multivariate patterns of electroencephalography (EEG) activity. (**A**) Decoding sensitivity in each stimulus condition across the epoch. Overall trends closely matched decoding accuracy. (**B**) Predicted (optimal sensitivity through maximum likelihood estimation [MLE] and aggregate A+V) and actual audiovisual sensitivity across the epoch. Coloured horizontal bars indicate cluster corrected periods where actual sensitivity significantly exceeded that which was predicted. Shaded error bars indicate ±1 SEM, n = 41.

The online version of this article includes the following figure supplement(s) for figure 4:

**Figure supplement 1.** Decoding sensitivity from frontal electrodes.

**Figure supplement 2.** Channel activity.

(*Figure 3B*). For auditory stimuli, information was primarily based over bilateral temporal regions, whereas for visual and audiovisual stimuli, the occipital electrodes carried the most information.

### Multivariate super-additivity

Although the univariate response did not show evidence for super-additivity, we expected the multivariate measure would be more sensitive to non-linear audiovisual integration. To test whether a super-additive interaction was present in the multivariate response, we calculated the sensitivity of the decoder in discriminating stimuli presented on the left and right side. The pattern of decoding sensitivity for auditory, visual, and audiovisual stimuli (*Figure 4A*) was similar to that in decoding accuracy (*Figure 3A*). Notably, audiovisual sensitivity was significantly greater than sensitivity to auditory and visual stimuli alone, particularly ~180 ms following stimulus onset. To test whether this enhanced

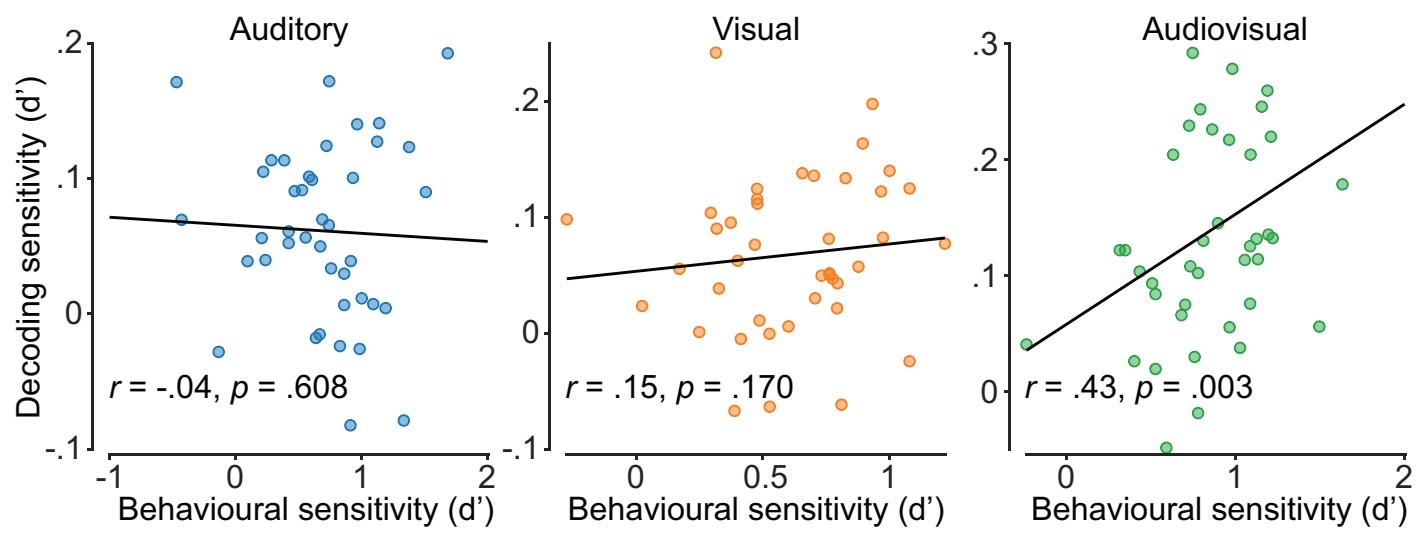

**Figure 5.** Audiovisual decoding sensitivity is significantly positively correlated with behavioural sensitivity. Correlations (Spearman's rho) are shown between decoding and behavioural sensitivity from the electroencephalography (EEG) session (150–250 ms post-stimulus onset) for each stimulus condition, with a line of best fit.

sensitivity reflected super-additivity, we compared decoding sensitivity for audiovisual stimuli with two estimates of linearly combined unisensory stimuli: (1) MLE predicted sensitivity based on auditory and visual sensitivity and (2) aggregate responses of auditory and visual stimuli (*Figure 4B*). We found that audiovisual sensitivity significantly exceeded both estimates of linear combination (MLE, ~160–220 ms post-stimulus; aggregate, ~150–250 ms). These results provide evidence of non-linear audiovisual integration in the multivariate pattern of EEG recordings. Taken together with the ERP results, our findings suggest that super-additive integration of audiovisual information is reflected in multivariate patterns of activity, but not univariate evoked responses.

## Neurobehavioural correlations

As behavioural and neural data violated assumptions of normality, we calculated rank-order correlations (Spearman's rho) between the average decoding sensitivity for each participant from 150 to 250 ms post-stimulus onset and behavioural performance on the EEG task. As Spearman's rho is resistant to outliers (*Wilcox, 2016*), we did not perform outlier rejection. We found that decoding sensitivity was significantly positively correlated with behavioural sensitivity for audiovisual stimuli ($r$=0.43, $p$=0.003), but not for auditory ($r$=–0.04, $p$=0.608) or visual stimuli ($r$=0.15, $p$=0.170) alone (*Figure 5*).

## Discussion

We tested for super-additivity in multivariate patterns of EEG responses to audiovisual stimuli. Participants judged the location of auditory, visual, and audiovisual stimuli while their brain activity was measured using EEG. As expected, participants' behavioural responses to audiovisual stimuli were more precise than those for unisensory auditory and visual stimuli. ERP analyses showed that although audiovisual stimuli elicited larger responses than visual stimuli, the overall response followed an additive principle. Critically, our multivariate analyses revealed that decoding sensitivity for audiovisual stimuli exceeded predictions of both MLE and aggregate auditory and visual information, indicating non-linear multisensory enhancement (i.e. super-additivity).

Participants localised audiovisual stimuli more accurately than unisensory in both the behavioural and EEG sessions. This behavioural facilitation in response to audiovisual stimuli is well established within the literature (*Bolognini et al., 2005*; *Frassinetti et al., 2002*; *Lovelace et al., 2003*; *Meredith and Stein, 1983*; *Senkowski et al., 2011*). In accordance with theories of optimal cue integration, we found participants' performance for audiovisual stimuli in both sessions matched that predicted by MLE (*Ernst and Banks, 2002*). Matching this 'optimal' prediction of performance indicates that

the auditory and visual cues were successfully integrated when presented together in the audiovisual condition (*Fetsch et al., 2013*).

Our EEG analyses revealed that for most spatial locations, audiovisual stimuli elicited a significantly greater neural response than exclusively visual stimuli approximately 80–120 ms after stimulus onset. Despite numerically larger ERPs to audiovisual than auditory stimuli, this effect failed to reach significance, most likely due to greater inter-trial variability in the auditory ERPs. Critically, however, the audiovisual ERPs consistently matched the sum of visual and auditory ERPs. Sub- or super-additive interaction effects in neural responses to multisensory stimuli are a hallmark of successful integration of unisensory cues in ERPs (*Besle et al., 2004*; *Stevenson et al., 2014*). An additive ERP in this context cannot imply successful multisensory integration, as the multisensory 'enhancement' may be the result of recording from distinct populations of unisensory neurons responding to the two unisensory sensory modalities (*Besle et al., 2009*). This invites the question of why we see evidence for integration at the behavioural level, but not in the amplitude of neural responses. One explanation could be that the signals measured by EEG simply do not contain evidence of non-linear integration because the super-additive responses are highly spatiotemporally localised and filtered out by the skull before reaching the EEG sensors. Another possibility, however, is that evidence for non-linear integration is only observable within the changing pattern of ERPs across sensors. Indeed, *Murray et al., 2016*, found that multisensory interactions followed from changes in scalp topography rather than net gains to ERP amplitude.

Our decoding results reveal that not only do audiovisual stimuli elicit more distinguishable patterns of activity than visual and auditory stimuli, but this enhancement exceeds that predicted by both optimal integration and the aggregate combination of auditory and visual responses. Critically, the non-linear enhancement of decoding sensitivity for audiovisual stimuli indicates the presence of an interactive effect for the integration of auditory and visual stimuli that was not evident from the univariate analyses. This indicates super-additive enhancement of the neural representation of integrated audiovisual cues and supports the interpretation that increased behavioural performance for multisensory stimuli is related to a facilitation of the neural response (*Fetsch et al., 2013*). This interaction was absent from univariate analyses (*Nikbakht et al., 2018*), suggesting that the neural facilitation of audiovisual processing is more nuanced than net increased excitation and may be associated with a complex pattern of excitatory and inhibitory neural activity, e.g., divisive normalization (*Ohshiro et al., 2017*).

The non-linear neural enhancement in decoding sensitivity for audiovisual stimuli occurred ~180 ms after stimulus onset, which is later than previously reported audiovisual interactions (<150 ms; *Cappe et al., 2010*; *Fort et al., 2002*; *Giard and Peronnet, 1999*; *Molholm et al., 2002*; *Murray et al., 2016*; *Senkowski et al., 2011*; *Talsma et al., 2007*; *Teder-Sälejärvi et al., 2002*). As stimulus characteristics and task requirements are likely to have a significant influence over the timing of multisensory interaction effects in EEG activity (*Calvert and Thesen, 2004*; *De Meo et al., 2015*), our use of peripheral spatial locations (where previous studies only used stimuli centrally) may explain the slightly later timing of our audiovisual effect. Indeed, our finding is consistent with previous multivariate studies which found that location information in EEG data, for both visual (*Rideaux, 2024*; *Robinson et al., 2021*) and auditory (*Bednar and Lalor, 2020*) stimuli, is maximal at ~190 ms following stimulus presentation.

An interesting aspect of our results is the apparent mismatch between the behavioural and neural responses. While the behavioural results meet the optimal statistical threshold predicted by MLE, the decoding analyses suggest that the neural response exceeds it. Though non-linear neural responses and statistically optimal behavioural responses are reliable phenomena in multisensory integration (*Alais and Burr, 2004*; *Ernst and Banks, 2002*; *Stanford and Stein, 2007*), the question remains – If neural super-additivity exists to improve behavioural performance, why is it not reflected in behavioural responses? A possible explanation for this neurobehavioural discrepancy is the large difference in timing between sensory processing and behavioural responses. A motor response would typically occur some time after the neural response to a sensory stimulus (e.g. 70–200 ms), with subsequent neural processes between perception and action that introduce noise (*Heekeren et al., 2008*) and may obscure super-additive perceptual sensitivity. In the current experiment, participants reported either the distribution of 20 serially presented stimuli (EEG session) or compared the positions of two stimuli (behavioural session), whereas the decoder attempted to recover the

location of every presented stimulus. While stimulus location could be represented with higher fidelity in multisensory relative to unisensory conditions, this would not necessarily result in better performance on a binary behavioural task in which multiple temporally separated stimuli are compared. One must also consider the inherent differences in how super-additivity is measured at the neural and behavioural levels. Neural super-additivity should manifest in responses to each individual stimulus. In contrast, behavioural super-additivity is often reported as proportion correct, which can only emerge between conditions after being averaged across multiple trials. The former is a biological phenomenon, while the latter is an analytical construct. In our experiment, we recorded neural responses for every presentation of a stimulus, but behavioural responses were only obtained after multiple stimulus presentations. Thus, the failure to find super-additivity in behavioural responses might be due to their operationalisation, with between-condition comparisons lacking sufficient sensitivity to detect super-additive sensory improvements. Future work should focus on experimental designs that can reveal super-additive responses in behaviour.

We also found a significant positive correlation between participants' behavioural judgements in the EEG session and decoding sensitivity for audiovisual stimuli. This result suggests that participants who were better at identifying stimulus location also had more reliably distinct patterns of neural activity. The lack of neurobehavioural correlation in the unisensory conditions might suggest a poor correspondence between the different tasks, perhaps indicative of the differences between behavioural and neural measures explained previously. However, multisensory stimuli have consistently been found to elicit stronger neural responses than unisensory stimuli (***Meredith and Stein, 1983***; ***Puce et al., 2007***; ***Senkowski et al., 2011***; ***Vroomen and Stekelenburg, 2010***), which has been associated with behavioural performance (***Frens and Van Opstal, 1998***; ***Wang et al., 2008***). Thus, the weaker signal-to-noise ratio in unisensory conditions may prevent correlations from being detected.

Any experimental design that varies stimulus location needs to consider the potential contribution of eye movements. We computed correlations between participants' average eye position and each stimulus position between the three sensory conditions (auditory, visual, and audiovisual; ***Figure 2—figure supplement 1***) and found evidence that participants made eye movements towards stimuli. A re-analysis of the data with a very strict eye-movement criterion (i.e. removing trials with eye movements >1.875°) revealed that the super-additive enhancement in decoding accuracy no longer survived cluster correction, suggesting that our results may be impacted by the consistent motor activity of saccades towards presented stimuli. Further investigation, however, suggests this is unlikely. Though the correlations were significantly different from 0, they were not significantly different from each other. If consistent saccades to audiovisual stimuli were responsible for the non-linear multisensory benefit we observed, we would expect to find a higher positive correlation between horizontal eye position and stimulus location in the audiovisual condition than in the auditory or visual conditions. Interestingly, eye movements corresponded more to stimulus location in the auditory and audiovisual conditions than in the visual condition, indicating that it was the presence of a sound, rather than a visual stimulus, that drove small eye movements. This could indicate that participants inadvertently moved their eyes when localising the origin of sounds. We also re-ran our analyses using the activity measured from the frontal electrodes alone (***Figure 4—figure supplement 1***). If the source of the non-linear decoding accuracy in the audiovisual condition was due to muscular activity produced by eye movements, there should be better decoding accuracy from sensors closer to the source. Instead, we found that decoding accuracy of stimulus location from the frontal electrodes (peak $d'$=0.08) was less than half that of decoding accuracy from the more posterior electrodes (peak $d'$=0.18). These results suggest that the source of neural activity containing information about stimulus position was located over occipito-parietal areas, consistent with our topographical analyses (inset of ***Figure 3***).

In summary, here, we have shown a non-linear enhancement in the neural representation of audiovisual stimuli relative to unisensory (visual/auditory) stimuli. This enhancement was obscured within univariate ERP analyses focusing exclusively on response amplitude but was revealed through inverted encoding analyses in feature space, suggesting that super-additive integration of audiovisual information is reflected within multivariate patterns of activity rather than univariate evoked responses. Further research on the multivariate representation of audiovisual integration may shed light on the neural mechanisms that facilitate this non-linear enhancement. In particular, future work may consider the influence of different stimulus features and task requirements on the timing and magnitude of the

audiovisual enhancement. How and when auditory and visual information are integrated to enhance multisensory processing remains an open question, with evidence for a complex combination of top-down and bottom-up interactions (*Delong and Noppeney, 2021*; *Keil and Senkowski, 2018*; *Rohe and Noppeney, 2018*). Our study highlights the importance of considering multivariate analyses in multisensory research, and the potential loss of stimulus-relevant neural information when relying solely on univariate responses.

## Methods

### Participants

Seventy-one human adults were recruited in return for payment. The study was approved by The University of Queensland Human Research Ethics Committee, and informed consent was obtained in all cases. Participants were first required to complete a behavioural session with above 60% accuracy in all conditions to qualify for the EEG session (see Behavioural session for details). Twenty-nine participants failed to meet this criterion and were excluded from further participation and analyses, along with one participant who failed to complete the EEG session with above-chance behavioural accuracy. This left a total of 41 participants (*M*=27.21 years; min = 20 years; max = 64 years; 24 females; 41 right-handed). Participants reported no neurological or psychiatric disorders and had normal visual acuity (assessed using a standard Snellen eye chart).

### Materials and procedure

The experiment was split into two separate sessions, with participants first completing a behavioural session followed by an EEG session. Each session had three conditions, in which the presented stimuli were either visual, auditory, or combined audio and visual (audiovisual). Conditions were not interleaved, but the order in which conditions were presented was counterbalanced across participants. Before each task, participants were given instructions and completed two rounds of practice for each condition.

### Apparatus

The experiment was conducted in a dark, acoustically and electromagnetically shielded room. For the EEG session, stimuli were presented on a 24-inch ViewPixx monitor (VPixx Technologies Inc, Saint-Bruno, QC, Canada) with 1920×1080-pixel resolution and a refresh rate of 144 Hz. Viewing distance was maintained at 54 cm using a chinrest. For the behavioural session, stimuli were presented on a 32-inch Cambridge Research Systems Display++ LCD monitor with a 1920×1080-pixel resolution, hardware gamma correction, and a refresh rate of 144 Hz. Viewing distance was maintained at 59.5 cm using a chinrest. Stimuli were generated in MATLAB v2021b (*The MathWorks Inc, 2021*) using the Psychophysics Toolbox (*Brainard, 1997*). Auditory stimuli were played through two loudspeakers placed either side of the display (80 cm apart for the behavioural session, 58 cm apart for the EEG session). The loudspeakers had a power handling capacity of 25–75 W and a nominal impedance of 6 Ω. In both experiments, an EyeLink 1000 Plus infrared eye tracker recorded gaze direction (SR Research Ltd, 2009) at a sampling rate of 1000 Hz.

### Stimuli

The EEG and behavioural paradigms used the same stimuli within each condition. Visual stimuli were Gaussian blobs (0.2 contrast, 16° diameter) presented for 16 ms on a mid-grey background. Auditory stimuli were 100 ms, 850 Hz tones with a decay function (sample rate = 44, 100 Hz; volume = 60 dBA SPL, as measured at the ears). Audiovisual stimuli were spatially and temporally matched combinations of the audio and visual stimuli, with no changes to stimuli properties. To manipulate spatial location, target stimuli were presented from multiple horizontal locations along the display, centred on linearly spaced locations from 15° visual angle to the left and right of the display centre (eight locations for behavioural, five for EEG). Auditory stimuli were played through two speakers placed equidistantly either side of the display. The perceived source location of auditory stimuli was manipulated via changes to interaural level and timing (*Whitworth and Jeffress, 1961*; *Wightman and Kistler, 1992*). The precise timing of when each speaker delivered an auditory stimulus was calculated from the following formula:

$$ITD_{L,R} = \frac{\sqrt{(x \pm r)^2 + z^2}}{s}$$

where $x$ and $z$ are the horizontal and forward distances in metres between the ears and the source of the sound on the display, respectively, $r$ is the head radius, and $s$ is the speed of sound. We used a constant approximate head radius of 8 cm for all participants. $r$ was added to $x$ for the left speaker and subtracted for the right speaker to produce the interaural time difference. For ±15° source locations, interaural timing difference was 1.7 ms. To simulate the decrease in sound intensity as a function of distance, we calculated interaural level differences for the left and right speakers by dividing the sounds by the left and right distance vectors. Finally, we resampled the sound using linear interpolation based on the calculations of the interaural level and timing differences. This process was used to calculate the soundwaves played by the left and right speakers for each of the possible stimulus locations on the display. The maximum interaural level difference between speakers was 0.14 $A$ for ±15° auditory locations, and 0.07 $A$ for ±7.5°.

## Behavioural session

During pilot testing, stimulus features (size, luminance, volume, frequency, etc.) were manipulated to make visual and auditory stimuli similarly difficult to spatially localise. These values were held constant in the main experiment. We employed a two-interval forced choice design to measure participants' audiovisual localisation sensitivity. Participants were presented with two consecutive stimuli and tasked with indicating, via button press, whether the first ('1' number-pad key) or second ('2' number-pad key) interval contained the more leftward stimulus. Each trial consisted of a central reference stimulus and a target stimulus presented at one of eight locations along the horizontal azimuth on the display. The presentation order of the reference and target stimuli was randomised across trials. Stimulus modality was auditory, visual, or audiovisual, presented in separate blocks with short breaks (~2 min) between conditions (see *Figure 6A* for an example trial). The order of conditions was counterbalanced across participants. Each condition consisted of 384 target presentations across the eight origin locations, leading to 48 presentations at each location.

## EEG session

In this session, the experimental task was changed slightly from the behavioural session to increase the number of stimulus presentations required for inverted encoding analyses of EEG data. Participants viewed and/or listened to a sequence of 20 stimuli, each of which was presented at one of five horizontal locations along the display (selected at random). At the end of each sequence, participants were tasked with indicating, via button press, whether more presentations appeared on the right ('right' arrow key) or the left ('left' arrow key) of the display. To minimise eye movements, participants were asked to fixate on a black dot presented 8° above the display centre (see *Figure 6B* for an example trial). The task used in the EEG session included the same blocked conditions as in the behavioural session, i.e., visual, auditory, and (congruent) audiovisual stimuli. As the locations of stimuli were selected at random, some sequences had an equal number of presentations on each side of the display, and thus had no correct 'left' or 'right' response; these trials were not included in the analysis of behavioural performance. Each block consisted of 10 trials, followed by a feedback display indicating the number of trials participants answered correctly. Each condition consisted of 12 blocks, yielding a total of 2400 presentations for each.

## EEG data preprocessing

EEG data were recorded using a 64-channel BioSemi system at a sampling rate of 1024 Hz, which was down-sampled to 512 Hz during preprocessing. Signals were recorded with reference to the CMS/DRL electrode loop, with bipolar electrodes placed above and below the eye, at the temples, and at each mastoid to monitor for eye movements and muscle artefacts. EEG preprocessing was undertaken in MATLAB using custom scripts and the EEGLAB toolbox (*Delorme and Makeig, 2004*; *Delorme and Makeig, 2026*; available at https://github.com/sccn/eeglab). Data were high-pass filtered at 0.25 Hz to remove baseline drifts and re-referenced according to the average of all 64 channels. Analyses were stimulus locked, with ERP responses segmented into 600 ms epochs from 100 ms before stimulus presentation to 500 ms after stimulus presentation. We removed trials with substantial

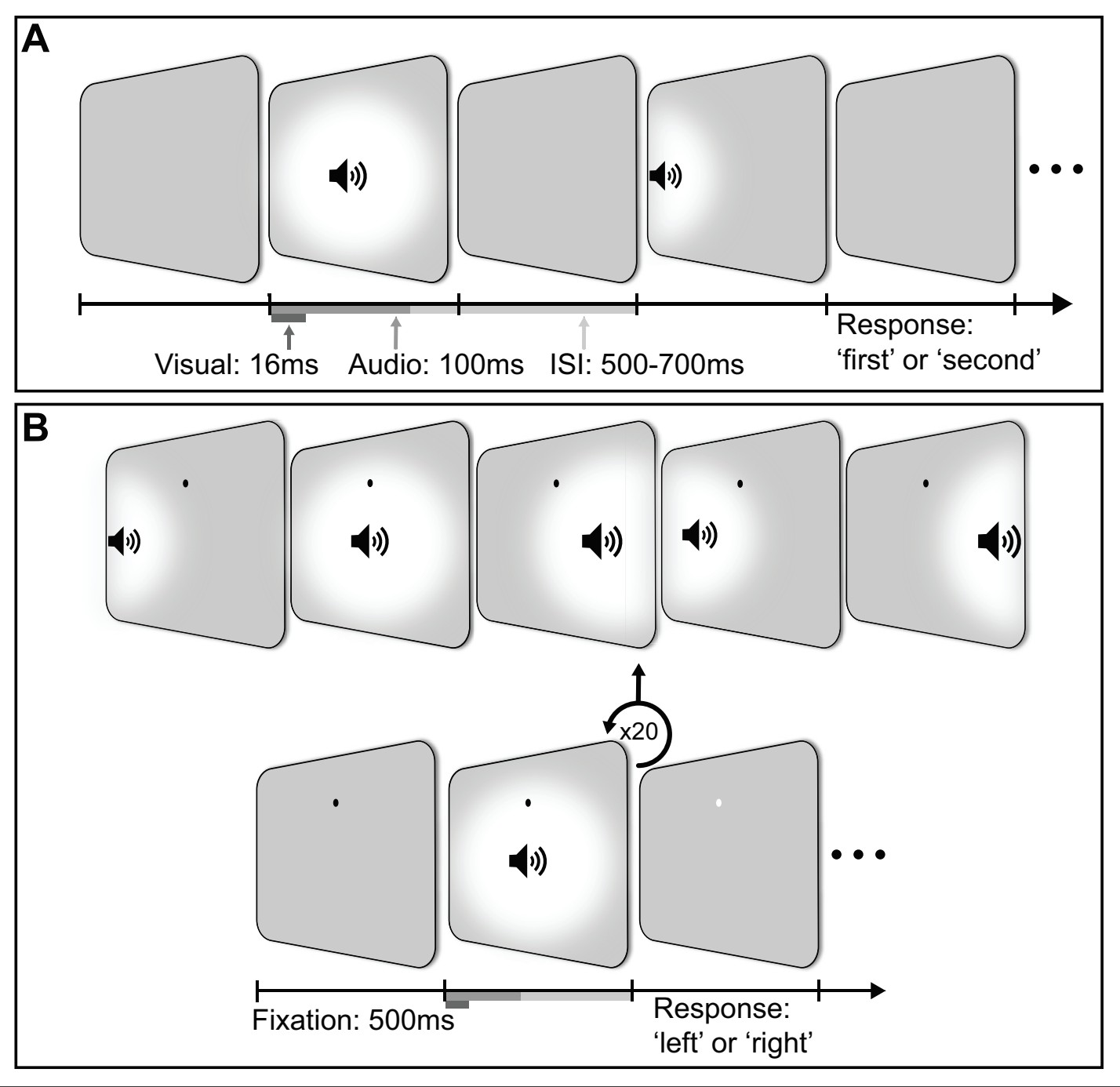

**Figure 6.** Experimental design of behavioural and electroencephalography (EEG) sessions. (**A**) An example trial for the audiovisual condition in the behavioural session. Each trial consisted of a (centred) reference stimulus and a target stimulus presented at one of eight locations along the horizontal meridian of the display. (**B**) An example trial for the audiovisual condition in the EEG session. The top row displays the possible locations of stimuli. In each trial, participants were presented with 20 stimuli that were each spatially localised to one of five possible locations along the horizontal meridian. The task was to determine if there were more stimuli presented to the left or right of fixation.

eye movements (>3.75 away from fixation) from the analyses. After the removal of eye movements, on average 2365 (SD = 56.94), 2346 (SD = 152.87), and 2350 (SD = 132.47) trials remained for auditory, visual, and audiovisual conditions, respectively, from the original 2400 per condition.

## Forward model

To describe the neural representations of sensory stimuli, we used an inverted modelling approach to reconstruct the location of stimuli based upon recorded ERPs (*Brouwer and Heeger, 2011*; *Harrison et al., 2023*). Analyses were performed separately for visual, auditory, and audiovisual stimuli. We first created an encoding model that characterised the patterns of activity in the EEG electrodes given the five locations of the presented stimuli. The encoding model was then used to obtain the inverse decoding model that described the transformation from electrode activity to stimulus location. We used a 10-fold cross-validation approach where 90% of the data were used to obtain the inverse model on which the remaining 10% of the data were decoded. Cross-validation was repeated 10 times such that all the data were decoded. For the purposes of these analyses, we assume that EEG electrode noise is isotropic across locations and additive with the signal.

Prior to the neural decoding analyses, we established the sensors that contained the most location information by treating time as the decoding dimension and obtaining the inverse models from each electrode, using 10-fold cross-validation. This revealed that location was primarily represented in posterior electrodes for visual and audiovisual stimuli, and in central electrodes for auditory stimuli. Thus, for all subsequent analyses, we only included signals from the central-temporal, parietal-occipital, occipital, and inion sensors for computing the inverse model (see final inset of *Figure 2*). As the model was fitted for multiple electrodes, subtle patterns of neural information contained within combinations of sensors could be detected.

The encoding model contained five hypothetical channels, with evenly distributed idealised location preferences between –15° and +15° viewing angles to the left and right of the display centre. Each channel consisted of a half-wave rectified sinusoid raised to the fifth power. The channels were arranged such that an idealised tuning curve of each location preference could be expressed as a weighted sum of the five channels. The observed activity for each presentation can be described by the following linear model:

$$\mathbf{B} = \mathbf{WC} + \mathbf{E}$$

where **B** indicates the EEG data (*m* electrodes × *n* presentations), **W** is a weight matrix (*m* electrodes × 5 channels) that describes the transformation from EEG activity to stimulus location, **C** denotes the hypothesised channel activities (5 channels × *n* presentations), and **E** indicates the residual errors.

To compute the inverse model, we estimated the weights that, when applied to the data, would reconstruct the channel activities with the least error. Due to the correlation between neighbouring electrodes, we took noise covariance into account when computing the model to optimise it for EEG data (*Harrison et al., 2023*; *Kok et al., 2017*; *Rideaux et al., 2023*). We then used the inverse model to reconstruct the stimulus location from the recorded ERP responses.

To assess how well the forward model captured location information in the neural signal per modality, two measures of performance were analysed. First, decoding accuracy was calculated as the similarity of the decoded location to the presented location, represented in arbitrary units. To test whether a super-additive interaction was present in the multivariate response, an additive threshold against which to compare the audiovisual response was required. However, it is unclear how the arbitrary units used to represent decoding accuracy translate to a measure of the linear summation of auditory and visual accuracy. As used for the behavioural analyses, MLE provides a framework for calculating the estimated optimal sensitivity of the combination of two sensory signals, according to signal detection theory principles. To compute decoding sensitivity (*d′*), required to apply MLE, we omitted trials where stimuli appeared in the centre of the display. The decoder's reconstructions of stimulus location were grouped for stimuli appearing on the left and right side of the display, respectively. The proportion of hits and misses was derived by comparing the decoded side to the presented side, which was then used to calculate *d′* for each condition (*Stanislaw and Todorov, 1999*). The *d′* from the auditory and visual conditions can be used to estimate the predicted 'optimal' sensitivity to audiovisual signals as calculated through the following formula:

$$d'AV = \sqrt{(d'A)^2 + (d'V)^2}$$

We can then compare actual audiovisual sensitivity to this auditory+visual sensitivity and test for super-additivity in the audiovisual condition as evidenced by the presence of a non-linear combination of auditory and visual stimuli. A similar method was previously employed to investigate depth estimation from motion and binocular disparity cues, decoded from BOLD responses (*Ban et al., 2012*).

To represent an additional 'additive' multivariate signal with which to compare the decoding sensitivity derived through MLE, we first matched the EEG data between unisensory conditions such that the order of presented stimulus locations was the same for the auditory and visual conditions. The auditory and visual condition data were then concatenated across sensors, and inverted encoding analyses were performed on the resulting 'additive' audiovisual dataset. This additive condition was designed to represent neural activity evoked by both the auditory and visual conditions, without any non-linear neural interaction, and served as a baseline for the audiovisual condition.

## Statistical analyses

Statistical analyses were performed in MATLAB v2021b. Two metrics of accuracy were calculated to assess behavioural performance. For the behavioural session, we calculated participants' sensitivity separately for each modality condition by fitting psychometric functions to the proportion of right-ward responses per stimulus location. In the EEG session, participants responded to multiple stimuli rather than individual presentations, so behavioural performance was assessed via $d'$. We derived $d'$ in each condition from the average proportion of hits and misses for each participant's performance in discriminating the side of the display on which more stimuli were presented (*Stanislaw and Todorov, 1999*). A one-sample Kolmogorov-Smirnov test for each condition revealed all conditions in both sessions violated assumptions of normality. A non-parametric two-sided Wilcoxon signed-rank test was therefore used to test for significant differences in behavioural accuracy between all conditions.

For the neural data, univariate ERPs were calculated by averaging EEG activity across presentations and channels for each stimulus location from −100 to 500 ms around stimulus onset. A conservative mass-based cluster correction was applied to account for spurious differences across time (*Pernet et al., 2015*). To test for significant differences between conditions, a paired-samples $t$-test was conducted between each condition at each time point. A one-sample $t$-test was used when comparing decoding accuracy against chance (i.e. zero). Next, the summed value of computed $t$ statistics associated with each comparison (separately for positive and negative values) was calculated within contiguous temporal clusters of significant values. We then simulated the null distribution of the maximum summed cluster values using permutation ($n$=5000) of the location labels, from which we derived the 95% percentile threshold value. Clusters identified in the data with a summed effect size value less than the threshold were considered spurious and removed.

## Acknowledgements

We thank R West for data collection and D Lloyd for technical assistance. This work was supported by the Australian Research Council (ARC) Discovery Early Career Researcher Awards awarded to RR (DE210100790) and AKR (DE200101159). RR was also supported by a National Health and Medical Research Council (Australia) Investigator Grant (2026318).

## Additional information

### Funding

| Funder | Grant reference number | Author |
| --- | --- | --- |
| Australian Research Council | DE210100790 | Reuben Rideaux |
| Australian Research Council | DE200101159 | Amanda K Robinson |

| Funder | Grant reference number | Author |
|---|---|---|
| National Health and Medical Research Council | 2026318 | Reuben Rideaux |

The funders had no role in study design, data collection and interpretation, or the decision to submit the work for publication.

## Author contributions

Zak Buhmann, Conceptualization, Resources, Data curation, Software, Formal analysis, Validation, Investigation, Visualization, Methodology, Writing – original draft, Project administration, Writing – review and editing; Amanda K Robinson, Conceptualization, Supervision, Funding acquisition, Methodology, Project administration, Writing – review and editing; Jason B Mattingley, Conceptualization, Resources, Supervision, Methodology, Project administration, Writing – review and editing; Reuben Rideaux, Conceptualization, Resources, Software, Formal analysis, Supervision, Funding acquisition, Visualization, Methodology, Project administration, Writing – review and editing

## Author ORCIDs

Zak Buhmann (iD) https://orcid.org/0009-0002-4249-462X
Amanda K Robinson (iD) https://orcid.org/0000-0002-7378-2803
Jason B Mattingley (iD) https://orcid.org/0000-0003-0929-9216
Reuben Rideaux (iD) https://orcid.org/0000-0001-8416-005X

## Ethics

Human subjects: All subjects were given detailed instructions of the task and what their consent means prior to the commencement of the experiment, and before they were invited to sign a consent form for the use of their anonymised data in publication. The experiment was approved by the Medicine LNR ethics committee to meet the requirements of the National Statement on Ethical Conduct in Human Research (2007, current revision), project number 2020/HE003101.

Reviewer #1 (Public review): https://doi.org/10.7554/eLife.97230.3.sa1
Reviewer #2 (Public review): https://doi.org/10.7554/eLife.97230.3.sa2
Author response https://doi.org/10.7554/eLife.97230.3.sa3

# Additional files

## Supplementary files
MDAR checklist

## Data availability
The behavioural and EEG data, and the scripts used for analysis and figure creation, are available at https://doi.org/10.17605/OSF.IO/8CDRA.

The following dataset was generated:

| Author(s) | Year | Dataset title | Dataset URL | Database and Identifier |
|---|---|---|---|---|
| Buhmann Z | 2026 | Inverted AV | https://doi.org/10.17605/OSF.IO/8CDRA | Open Science Framework, 10.17605/OSF.IO/8CDRA |

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
