## [Editor Report · eLife Assessment]

Despite the well-established facilitatory effects of multisensory integration on behavioural measures, standard neuroimaging approaches have yet to reliably and precisely identify the corresponding neural correlates. In this **valuable** paper, Buhmann et al. leverage EEG decoding methods, moving beyond traditional univariate analyses, to capture these correlates. They present **solid** evidence that this approach can effectively estimate multisensory integration in humans across a broad range of contexts.

---

## [Referee Report · Reviewer #1 (Public review)]

This study presents a novel application of inverted encoding (i.e., decoding) to detect non-linear correlates of crossmodal integration in human neural activity, using EEG (electroencephalography). The method is successfully applied to data from a group of 41 participants, performing a spatial localization task on auditory, visual, and audio-visual events. The analyses clearly show a behavioural superiority for audio-visual localization. Like previous studies, the results when using traditional univariate ERP analyses were inconclusive, showing once more the need for alternative, more sophisticated approaches. The inverted encoding approach of this study, harnessing on the multivariate nature of the signal, captured clear signs of super-additive responses, considered by many as the hallmark of multisensory integration. Despite the removal of eye-movement artefacts from the signal eliminated the significant decoding effect, the author's control analyses showed that decoding is more effective from parietal, compared to frontal electrodes, thereby ruling out ocular contamination as the sole origin of the relevant signal.

This significant finding can bear important advances in the many fields where multisensory integration has been shown to play an important role, by providing a way to bring much needed coherence across levels of analysis, from behaviour to single-cell electrophysiology. To achieve this, it would be ideal to contrast whether the pattern of super-additive effects in other scenarios where clear behavioural signs of multisensory integration are also observed. One could also try to further support the posterior origin of the super-additive effects by source localization.

Comments on revised version:

All my previous concerns have been addressed. I congratulate the authors on a very nice paper.

---

## [Referee Report · Reviewer #2 (Public review)]

Summary:

This manuscript seeks to reconcile observations in multisensory perception - from behavior and from neural responses. It is intuitively obvious that perceiving a stimulus via two senses results in better performance than one alone. However, the nature of this interaction is complicated and relating different measures (behavioural, neural) is challenging.

It is not uncommon to observe that for a perceptual task the percentage of correct responses seen with two senses is higher than the sum of the percentage correct obtained with each modality individually. i.e. the gains are "superadditive". The gains of adding a second sense are typically larger when the performance with the first sense is relatively poor - this effect is often called the principle inverse effectiveness. More generally, what this tells us is that performance in a multisensory perceptual task is a non-linear sum of performance for each sensory modality alone. In invasive recordings from single neurons, a wide range of non-linear interactions is observed - some superadditive, and some sub-additive.

Despite this abundance evidence of non-linearity in some measures of multisensory integration, evoked responses (EEG) to such sensory stimuli often show little evidence of it - and this is the problem this manuscript tackles. The key assertion made is that a univariate analysis of the EEG signal is likely to average out non-linear effects of integration. This is a reasonable assertion, and their analysis does indeed provide evidence that a multivariate approach can reveal non-linear interactions in the evoked responses.

Strengths:

It is of great value to understand how the process of multisensory integration occurs, and despite a wealth of observations of the benefits of perceiving the world with multiple senses, we still lack a reasonable understanding of how the brain integrates information. For example - what underlies the large individual differences in the benefits of two senses over one? One way to tackle this is via brain imaging, but this is problematic if important features of the processing - such as non-linear interactions are obscured by the lack of specificity of the measurements. The approach they take to analysis of the EEG data allows the authors to look in more detail at the variation in activity across EEG electrodes, which averaging across electrodes cannot.

This version of the manuscript is well written and for the most part clear and the report of non-linear summation of neural responses is convincing. A particular strength of the paper is their use of a statistical model of multisensory integration as their "null" model of neural responses, and the "inverted-encoder" which infers an internal representation of the stimulus which can explain the EEG responses. This encoder generates a prediction of decoding performance, which can be used to generate predictions of multisensory decoding from unisensory decoding, or from a sum of the unisensory internal representations.

In behavioural performance, it is frequently observed that the performance increase from two senses is close to what is expected from the optimal integration of information across the senses, in a statistical sense. It can be plausibly explained by assuming that people are able to weight sensory inputs according to their reliability - and somewhat optimally. Critically the apparent "superadditive" effect on performance described above does not require any non-linearity in the sum of information across the senses, but can arise from correctly weighting the information according to reliability.

The authors apply a similar model to predict the neural responses expected to audiovisual stimuli from the neural responses to audio and visual stimuli alone, assuming optimal statistical integration of information. The neural responses to audiovisual stimuli exceed the predictions of this model and this is the main evidence supporting their conclusion, and it is convincing.

Weaknesses:

The main weakness of the manuscript is that their behavioural data show no evidence of performance that exceeds the predictions of these statistical models. In fact, the models predict multisensory performance from unisensory performance pretty well. This makes it hard to interpret their results, as surely if these nonlinear neural interactions underlie the behaviour, then we should be able to see evidence of it in the behaviour. I cannot offer an easy explanation for this.

Overall, therefore, I applaud the motivation and the sophistication of the analysis method and think it shows great promise for tackling these problems.

---

## [Author Response]

The following is the authors’ response to the original reviews.

**Reviewer 1:**

We thank Reviewer 1 for their helpful comments and hope that the changes made to the revised manuscript have addressed their points.

This study presents a novel application of the inverted encoding (i.e., decoding) approach to detect the correlates of crossmodal integration in the human EEG (electrophysiological) signal. The method is successfully applied to data from a group of 41 participants, performing a spatial localization task on auditory, visual, and audiovisual events. The analyses clearly show a behavioural superiority for audio-visual localization. Like previous studies, the results when using traditional univariate ERP analyses were inconclusive, showing once more the need for alternative, more sophisticated approaches. Instead, the principal approach of this study, harnessing the multivariate nature of the signal, captured clear signs of super-additive responses, considered by many as the hallmark of multisensory integration. Unfortunately, the manuscript lacks many important details in the descriptions of the methodology and analytical pipeline. Although some of these details can eventually be retrieved from the scripts that accompany this paper, the main text should be self-contained and sufficient to gain a clear understanding of what was done. (A list of some of these is included in the comments to the authors). Nevertheless, I believe the main weakness of this work is that the positive results obtained and reported in the results section are conditioned upon eye movements. When artifacts due to eye movements are removed, then the outcomes are no longer significant.Therefore, whether the authors finally achieved the aims and showed that this method of analysis is truly a reliable way to assess crossmodal integration, does not stand on firm ground. The worst-case scenario is that the results are entirely accounted for by patterns of eye movements in the different conditions. In the best-case scenario, the method might truly work, but further experiments (and/or analyses) would be required to confirm the claims in a conclusive fashion.One first step toward this goal would be, perhaps, to facilitate the understanding of results in context by reporting both the uncorrected and corrected analyses in the main results section. Second, one could try to support the argument given in the discussion, pointing out the origin of the super-additive effects in posterior electrode sites, by also modelling frontal electrode clusters and showing they aren't informative as to the effect of interest.

We performed several additional analyses to address concerns that our main result was caused by different eye movement patterns between conditions. We re-ran our key analyses using activity exclusively from frontal electrodes, which revealed poorer decoding performance than that from posterior electrodes. If eye movements were driving the non-linear enhancement in the audiovisual condition, we would expect stronger decoding using sensors closer to the source, i.e., the extraocular muscles. We also computed the correlations between average eye position and stimulus position for each condition to evaluate whether participants made larger eye movements in the audiovisual condition, which might have contributed to better decoding results. Though we did find evidence for eye movements toward stimuli, the degree of movement did not significantly differ between conditions.

Furthermore, we note that the analysis using a stricter eye movement criterion, acknowledged in the Discussion section of the original manuscript, resulted in very similar results to the original analysis. There was significantly better decoding in the AV condition (as measured by d') than the MLE prediction, but this difference did not survive cluster correction. The most likely explanation for this is that the strict eye movement criterion combined with our conservative measure of (mass-based) cluster correction led to reduced power to detect true differences between conditions. Taken together with the additional analyses described in the revised manuscript and supplementary materials, the results show that eye movements are unlikely to account for differences between the multisensory and unisensory conditions. Instead, our decoding results likely reflect nonlinear neural integration between audio and visual sensory information.

“Any experimental design that varies stimulus location needs to consider the potential contribution of eye movements. We computed correlations between participants’ average eye position and each stimulus position between the three sensory conditions (auditory, visual and audiovisual; Figure S1) and found evidence that participants made eye movements toward stimuli. A re-analysis of the data with a very strict eye-movement criterion (i.e., removing trials with eye movements >1.875º) revealed that the super-additive enhancement in decoding accuracy no longer survived cluster correction, suggesting that our results may be impacted by the consistent motor activity of saccades towards presented stimuli. Further investigation, however, suggests this is unlikely. Though the correlations were significantly different from 0, they were not significantly different from each other. If consistent saccades to audiovisual stimuli were responsible for the nonlinear multisensory benefit we observed, we would expect to find a higher positive correlation between horizontal eye position and stimulus location in the audiovisual condition than in the auditory or visual conditions. Interestingly, eye movements corresponded more to stimulus location in the auditory and audiovisual conditions than in the visual condition, indicating that it was the presence of a sound, rather than a visual stimulus, that drove small eye movements. This could indicate that participants inadvertently moved their eyes when localising the origin of sounds. We also re-ran our analyses using the activity measured from the frontal electrodes alone (Figure S2). If the source of the nonlinear decoding accuracy in the audiovisual condition was due to muscular activity produced by eye movements, there should be better decoding accuracy from sensors closer to the source. Instead, we found that decoding accuracy of stimulus location from the frontal electrodes (peak d' = 0.08) was less than half that of decoding accuracy from the more posterior electrodes (peak d' = 0.18). These results suggest that the source of neural activity containing information about stimulus position was located over occipito-parietal areas, consistent with our topographical analyses (inset of Figure 3).”

The univariate ERP analyses an outdated contrast, AV <> A + V to capture multisensory integration. A number of authors have pointed out the potential problem of double baseline subtraction when using this contrast, and have recommended a number of solutions, experimental and analytical. See for example: [1] and [2].(1) Teder-Salejarvi, W. A., McDonald, J. J., Di Russo, F., & Hillyard, S. A. (2002). Cognitive Brain Research, 14, 106-114.(2) Talsma, D., & Woldorff, M. G. (2005). Journal of cognitive neuroscience, 17(7), 1098-1114.

We thank the reviewer for raising this point. Comparing ERPs across different sensory conditions requires careful analytic choices to discern genuine sensory interactions within the signal. The AV <> (A +V) contrast has often been used to detect multisensory integration, though any non-signal related activity (i.e. anticipatory waves; Taslma & Woldorff, 2005) or pre-processing manipulation (e.g. baseline subtraction; Teder-Sälejärvi et al., 2002) will be doubled in (A + V) but not in AV. Critically, we did not apply a baseline correction during preprocessing and thus our results are not at risk of double-baseline subtraction in (A + V). Additionally, we temporally jittered the presentation of our stimuli to mitigate the potential influence of consistent overlapping ERP waves (Talsma & Woldorff, 2005).

The results section should provide the neurometric curve/s used to extract the slopes of the sensitivity plot (Figure 2B).

We thank the reviewer for raising this point of clarification. The sensitivity plots for Figures 2B and 2C were extracted from the behavioural performance of the behavioural and EEG tasks, respectively. The sensitivity plot for Figure 2B was extracted from individual psychometric curves, whereas the d’ values for Figure 2C were calculated from the behavioural data for the EEG task. This information has been clarified in the manuscript.

“Figure 1. Behavioural performance is improved for audiovisual stimuli. (A) Average accuracy of responses across participants in the behavioural session at each stimulus location for each stimulus condition, fitted to a psychometric curve. Steeper curves indicate greater sensitivity in identifying stimulus location. (B) Average sensitivity across participants in the behavioural task, estimated from psychometric curves, for each stimulus condition. The red cross indicates estimated performance assuming optimal (MLE) integration of unisensory cues. (C) Average behavioural sensitivity across participants in the EEG session for each stimulus condition. Error bars indicate ±1 SEM.”

The encoding model was fitted for each electrode individually; I wonder if important information contained as combinations of (individually non-significant) electrodes was then lost in this process and if the authors consider that this is relevant.

Although the encoding model was fitted for each electrode individually for the topographic maps (Figure 4B), in all other analyses the encoding model was fitted across a selection of electrodes (see final inset of Figure 3). As this electrode set was used for all other neural analyses, our model would allow for the detection of important information contained in the neural patterns across electrodes. This information has been clarified in the manuscript.

“Thus, for all subsequent analyses we only included signals from the central-temporal, parietal-occipital, occipital and inion sensors for computing the inverse model (see final inset of Figure 2). As the model was fitted for multiple electrodes, subtle patterns of neural information contained within combinations of sensors could be detected.”

Neurobehavioral correlations could benefit from outlier rejection and the use of robust correlation statistics.

We thank the reviewer for raising this issue. Note, however, that the correlations we report are resistant to the influence of outliers because we used Spearman’s rho1 (as opposed to Pearson’s). This information has been communicated in the manuscript.

(1) Wilcox, R.R. (2016), Comparing dependent robust correlations. *British Journal of Mathematical & Statistical Psychology, 69*(3), 215-224. https://doi.org/10.1111/bmsp.12069

*“Neurobehavioural correlations.* As behavioural and neural data violated assumptions of normality, we calculated rank-order correlations (Spearman’s rho) between the average decoding sensitivity for each participant from 150-250 ms poststimulus onset and behavioural performance on the EEG task. As Spearman’s rho is resistant to outliers (Wilcox, 2016), we did not perform outlier rejection.”

“Wilcox, R.R. (2016), Comparing dependent robust correlations. *British Journal of Mathematical & Statistical Psychology, 69*(3), 215-224. https://doi.org/10.1111/bmsp.12069”

Many details that are important for the reader to evaluate the evidence and to understand the methods and analyses aren't given; this is a non-exhaustive list:

We thank the reviewer for highlighting these missing details. We have updated the manuscript where necessary to ensure the methods and analyses are fully detailed and replicable.

- specific parameters of the stimuli and performance levels. Just saying "similarly difficult" or "marginally higher volume" is not enough to understand exactly what was done.

“The perceived source location of auditory stimuli was manipulated via changes to interaural level and timing (Whitworth & Jeffress, 1961; Wightman & Kistler, 1992). The precise timing of when each speaker delivered an auditory stimulus was calculated from the following formula:\begin{document}$$\displaystyle I T D_{L, R}=\frac{\sqrt{(x \pm r)^{2}+z^{2}}}{s}$$\end{document}

where *x* and *z* are the horizontal and forward distances in metres between the ears and the source of the sound on the display, respectively, *r* is the head radius, and *s* is the speed of sound. We used a constant approximate head radius of 8 cm for all participants. *r* was added to *x* for the left speaker and subtracted for the right speaker to produce the interaural time difference. For ±15° source locations, interaural timing difference was 1.7 ms. To simulate the decrease in sound intensity as a function of distance, we calculated interaural level differences for the left and right speakers by dividing the sounds by the left and right distance vectors. Finally, we resampled the sound using linear interpolation based on the calculations of the interaural level and timing differences. This process was used to calculate the soundwaves played by the left and right speakers for each of the possible stimulus locations on the display. The maximum interaural level difference between speakers was 0.14 *A* for ±15° auditory locations, and 0.07 *A* for ±7.5°.”

- where are stimulus parameters adjusted individually or as a group? Which method was followed?

To clarify, stimulus parameters (frequency, size, luminance, volume, location, etc.) were manipulated throughout pilot testing *only*. Parameters were adjusted to achieve similar pilot behavioural results between the auditory and visual conditions. For the experiment proper, parameters remained constant for both tasks and were the same for all participants.

“During pilot testing, stimulus features (size, luminance, volume, frequency etc.) were manipulated to make visual and auditory stimuli similarly difficult to spatially localize. These values were held constant in the main experiment.”

- specify which response buttons were used.

“Participants were presented with two consecutive stimuli and tasked with indicating, via button press, whether the first (‘1’ number-pad key) or second (‘2’ number-pad key) interval contained the more leftward stimulus.”

“At the end of each sequence, participants were tasked with indicating, via button press, whether more presentations appeared on the right (‘right’ arrow key) or the left (‘left’ arrow key) of the display.”

- no information is given as to how many trials per condition remained on average, for analysis.

The average number of remaining trials per condition after eye-movement analysis is now included in the Methods section of the revised manuscript.

“We removed trials with substantial eye movements (>3.75 away from fixation) from the analyses. After the removal of eye movements, on average 2365 (*SD* = 56.94), 2346 (*SD* = 152.87) and 2350 (*SD* = 132.47) trials remained for auditory, visual and audiovisual conditions, respectively, from the original 2400 per condition.”

- no information is given on the specifics of participant exclusion criteria. (even if the attrition rate was surprisingly high, for such an easy task).

The behavioural session also served as a screening task. Although the task instructions were straightforward, perceptual discrimination was not easy due to the ambiguity of the stimuli. Auditory localization is not very precise, and the visual stimuli were brief, dim, and diffuse. The behavioural results reflect the difficulty of the task. Attrition rate was high as participants who scored below 60% correct in any condition were deemed unable to accurately perform the task, were not invited to complete the subsequent EEG session, and omitted from the analyses. We have included the specific criteria in the manuscript.

“Participants were first required to complete a behavioural session with above 60% accuracy in all conditions to qualify for the EEG session (see *Behavioural session* for details).”

- EEG pre-processing: what filter was used? How was artifact rejection done? (no parameters are reported); How were bad channels interpolated?

We used a 0.25 Hz high-pass filter to remove baseline drifts, but no low-pass filter. In line with recent studies on the undesirable influence of EEG preprocessing on ERPs1, we opted to avoid channel interpolation and artifact rejection. This was erroneously reported in the manuscript and has now been clarified. For the sake of clarity, here we demonstrate that a reanalysis of data using channel interpolation and artifact rejection returned the same pattern of results.

(1) Delorme, A. (2023). EEG is better left alone. *Scientific Reports*, 13, 2372. https://doi.org/10.1038/s41598-023-27528-0

- specific electrode locations must be given or shown in a plot (just "primarily represented in posterior electrodes" is not sufficiently informative).

A diagram of the electrodes used in all analyses is included within Figure 3, and we have drawn readers’ attention to this in the revised manuscript.

“Thus, for all subsequent analyses we only included signals from the central-temporal, parietal-occipital, occipital and inion sensors for computing the inverse model (see final inset of Figure 2).”

- ERP analysis: which channels were used? What is the specific cluster correction method?

We used a conservative mass-based cluster correction from Pernet et al. (2015) - this information has been clarified in the manuscript.

“A conservative mass-based cluster correction was applied to account for spurious differences across time (Pernet et al., 2015).”

“Pernet, C. R., Latinus, M., Nichols, T. E., & Rousselet, G. A. (2015). Cluster-based computational methods for mass univariate analyses of event-related brain potentials/fields: A simulation study. *Journal of Neuroscience Methods*, *250*, 85-93. https://doi.org/10.1016/j.jneumeth.2014.08.003”

- results: descriptive stats on performance must be given (instead of saying "participants performed well").

The mean and standard deviation of participants’ performance for each condition in the behavioural and EEG experiments are now explicitly mentioned in the manuscript.

“A quantification of the behavioural sensitivity (i.e., steepness of the curves) revealed significantly higher sensitivity for the audiovisual stimuli (*M* = .04, *SD* = .02) than for the auditory stimuli alone (*M* = .03, *SD* = .01; Z = -3.09, *p* = .002), and than for the visual stimuli alone (*M* = .02, *SD* = .01; Z = -5.28, *p* = 1.288e-7; Figure 1B). Sensitivity for auditory stimuli was also significantly higher than sensitivity for visual stimuli (*Z* = 2.02, *p* = .044).”

“We found a similar pattern of results to those in the behavioural session; sensitivity for audiovisual stimuli (*M* = .85, *SD* = .33) was significantly higher than for auditory (*M* = .69, *SD* = .41; *Z* = -2.27, *p* = .023) and visual stimuli alone (*M* = .61, *SD* = .29; *Z* = -3.52, *p* = 4.345e-4), but not significantly different from the MLE prediction (*Z* = -1.07, *p* = .285).”

- sensitivity in the behavioural and EEG sessions is said to be different, but no comparison is given. It is not even the same stimulus set across the two tasks...

This relationship was noted as a potential explanation for the higher sensitivities obtained in the EEG task, and was not intended to stand up to statistical scrutiny. We agree it makes little sense to compare statistically between the EEG and behavioural results as they were obtained from different tasks. We would like to clarify, however, that the stimuli used in the two tasks were the same, with the exception that in the EEG task the stimuli were presented from 5 locations versus 8 in the behavioural task. To avoid potential confusion, we have removed the offending sentence from the manuscript:

**Reviewer 2:**
Their measure of neural responses is derived from the decoder responses, and this takes account of the reliability of the sensory representations - the d' statistics - which is an excellent thing. It also means if I understand their analysis correctly (it could bear clarifying - see below), that they can generate from it a prediction of the performance expected if an optimal decision is made combining the neural signals from the individual modalities. I believe this is the familiar root sum of squares d' calculation (or very similar). Their decoding of the audiovisual responses comfortably exceeds this prediction and forms part of the evidence for their claims.Yet, superadditivity - including that in evidence in the principle of inverse effectiveness more typically quantifies the excess over the sum of proportions correct in each modality. Their MLE d' statistic can already predict this form of superadditivity. Therefore, the superadditivity they report here is not the same form of superadditivity that is usually referred to in behavioural studies. It is in fact a stiffer definition. What their analysis tests is that decoding performance exceeds what would be expected from an optimally weighted linear integration of the unisensory information. As this is not the common definition it is difficult to relate to behavioral superadditivity reported in much literature (of percentage correct). This distinction is not at all clear from the manuscript.But the real puzzle is here: The behavioural data or this task do not exceed the optimal statistical decision predicted by signal detection theory (the MLE d'). Yet, the EEG data would suggest that the neural processing is exceeding it. So why, if the neural processing is there to yield better performance is it not reflected in the behaviour? I cannot explain this, but it strikes me that the behaviour and neural signals are for some reason not reflecting the same processing.Be explicit and discuss this mismatch they observe between behaviour and neural responses.

Thank you, we agree that it is worth expanding on the observed disconnect between MSI in behaviour and neural signals. We have included an additional paragraph in the Discussion of the revised manuscript. Despite the mismatch, we believe the behavioural and neural responses still reflect the same underlying processing, but at different levels of sensitivity. The behavioural result likely reflects a coarse down-sampling of the precision in location representation, and thus less likely to reflect subtle MSI enhancements.

“An interesting aspect of our results is the apparent mismatch between the behavioural and neural responses. While the behavioural results meet the optimal statistical threshold predicted by MLE, the decoding analyses suggest that the neural response exceeds it. Though non-linear neural responses and statistically optimal behavioural responses are reliable phenomena in multisensory integration (Alais & Burr, 2004; Ernst & Banks, 2002; Stanford & Stein, 2007), the question remains – if neural super-additivity exists to improve behavioural performance, why is it not reflected in behavioural responses? A possible explanation for this neurobehavioural discrepancy is the large difference in timing between sensory processing and behavioural responses. A motor response would typically occur some time after the neural response to a sensory stimulus (e.g., 70-200 ms), with subsequent neural processes between perception and action that introduce noise (Heekeren et al., 2008) and may obscure super-additive perceptual sensitivity. In the current experiment, participants reported either the distribution of 20 serially presented stimuli (EEG session) or compared the positions of two stimuli (behavioural session), whereas the decoder attempts to recover the location of every presented stimulus. While stimulus location could be represented with higher fidelity in multisensory relative to unisensory conditions, this would not necessarily result in better performance on a binary behavioural task in which multiple temporally separated stimuli are compared. One must also consider the inherent differences in how super-additivity is measured at the neural and behavioural levels. Neural super-additivity should manifest in responses to each individual stimulus. In contrast, behavioural super-additivity is often reported as proportion correct, which can only emerge between conditions after being averaged across multiple trials. The former is a biological phenomenon, while the latter is an analytical construct. In our experiment, we recorded neural responses for every presentation of a stimulus, but behavioural responses were only obtained after multiple stimulus presentations. Thus, the failure to find super-additivity in behavioural responses might be due to their operationalisation, with between-condition comparisons lacking sufficient sensitivity to detect super-additive sensory improvements. Future work should focus on experimental designs that can reveal super-additive responses in behaviour.”

Re-work the introduction to explain more clearly the relationship between the behavioural superadditivities they review, the MLE model, and the superadditivity it actually tests.

We agree it is worth discussing how super-additivity is operationalised across neural and behavioural measures. However, we do not believe the behavioural studies we reviewed claimed super-additive behavioural enhancements. While MLE is often used as a behavioural marker of successful integration, it is not necessarily used as evidence for super-additivity within the behavioural response, as it relies on linear operations.

“It is important to consider the differences in how super-additivity is classified between neural and behavioural measures. At the level of single neurons, superadditivity is defined as a non-linear response enhancement, with the multisensory response exceeding the sum of the unisensory responses. In behaviour, meanwhile, it has been observed that the performance improvement from combining two senses is close to what is expected from optimal integration of information across the senses (Alais & Burr, 2004; Stanford & Stein, 2007). Critically, behavioural enhancement of this kind does not require non-linearity in the neural response, but can arise from a reliability-weighted average of sensory information. In short, behavioural performance that conforms to MLE is not necessarily indicative of neural super-additivity, and the MLE model can be considered a linear baseline for multisensory integration.”

Regarding the auditory stimulus, this reviewer notes that interaural time differences are unlikely to survive free field presentation.

Despite the free field presentation, in both the pilot test and the study proper participants were able to localize auditory stimuli significantly above chance.

"However, other studies have found super-additive enhancements to the amplitude of sensory event-related potentials (ERPs) for audiovisual stimuli (Molholm et al., 2002; Talsma et al., 2007), especially when considering the influence of stimulus intensity (Senkowski et al., 2011)." - this makes it obvious that there are some studies which show superadditivity. It would have been good to provide a little more depth here - as to what distinguished those studies that reported positive effects from those that did not.

We have provided further detail on how super-additivity appears to manifest in neural measures.

“In EEG, meanwhile, the evoked response to an audiovisual stimulus typically conforms to a sub-additive principle (Cappe et al., 2010; Fort et al., 2002; Giard & Peronnet, 1999; Murray et al., 2016; Puce et al., 2007; Stekelenburg & Vroomen, 2007; Teder- Sälejärvi et al., 2002; Vroomen & Stekelenburg, 2010). However, when the principle of inverse effectiveness is considered and relatively weak stimuli are presented together, there has been some evidence for super-additive responses (Senkowski et al., 2011).”

“While behavioural outcomes for multisensory stimuli can be predicted by MLE, and single neuron responses follow the principles of inverse effectiveness and super- additivity, among others (Rideaux et al., 2021), how audiovisual super-additivity manifests within populations of neurons is comparatively unclear given the mixed findings from relevant fMRI and EEG studies. This uncertainty may be due to biophysical limitations of human neuroimaging techniques, but it may also be related to the analytic approaches used to study these recordings. For instance, superadditive responses to audiovisual stimuli in EEG studies are often reported from very small electrode clusters (Molholm et al., 2002; Senkowski et al., 2011; Talsma et al., 2007), suggesting that neural super-additivity in humans may be highly specific. However, information encoded by the brain can be represented as increased activity in some areas, accompanied by decreased activity in others, so simplifying complex neural responses to the average rise and fall of activity in specific sensors may obscure relevant multivariate patterns of activity evoked by a stimulus.”

P9. "(25-75 W, 6 Ω)." This is not important, but it is a strange way to cite the power handling of a loudspeaker.

“The loudspeakers had a power handling capacity of 25-75 W and a nominal impedance of 6 Ω.”

I am struggling to understand the auditory stimulus:"Auditory stimuli were 100 ms clicks". Is this a 100-ms long train of clicks? A single pulse which is 100ms long would not sound like a click, but two clicks once filtered by the loudspeaker. Perhaps they mean 100us."..with a flat 850 Hz tone embedded within a decay envelope". Does this mean the tone is gated - i.e. turns on and off slowly? Or is it constant?

We thank the reviewer for catching this. ‘Click’ may not be the most apt way of defining the auditory stimulus. It was a 100 ms square wave tone with decay, i.e., with an onset at maximal volume before fading gradually. Given that the length of the stimulus was 100 ms, the decay occurs quickly and provides a more ‘click-like’ percept than a pure tone. We have provided a representation of the sound below for further clarification. This represents the amplitude from the L and R speakers for maximally-left and maximally-right stimuli. We have added this clarification in the revised manuscript.

“Auditory stimuli were 100 ms, 850 Hz tones with a decay function (sample rate = 44, 100 Hz; volume = 60 dBA SPL, as measured at the ears).”

P10. "Stimulus modality was either auditory, visual, or audiovisual. Trials were blocked with short (~2 min) breaks between conditions".Presumably the blocks were randomised across participants.

Condition order was not randomised across participants, but counterbalanced. This has been clarified in the manuscript.

“Stimulus modality was auditory, visual or audiovisual, presented in separate blocks with short breaks (~2 min) between conditions (see Figure 6A for an example trial). The order of conditions was counterbalanced across participants.”

P15. Feels like there is a step not described here: "The d' of the auditory and visual conditions can be used to estimate the predicted 'optimal' sensitivity of audiovisual signals as calculated through MLE." Do they mean sqrt [(d'A)^2 + (d'V)^2] ? If it is so simple then it may as well be made explicit here. A quick calculation from eyeballing Figures 2B and 2C suggests this is the case.

We thank the reviewer for raising this point of clarification. Yes, the ‘optimal’ audiovisual sensitivity was calculated as the hypotenuse of the auditory and visual sensitivities. This calculation has been made explicit in the revised manuscript.

The d’ from the auditory and visual conditions can be used to estimate the predicted ‘optimal’ sensitivity to audiovisual signals as calculated through the following formula:\begin{document}$$\displaystyle d^{\prime} A V=\sqrt{\left(d^{\prime} A\right)^{2}+\left(d^{\prime} V\right)^{2}}$$\end{document}

"The perceived source location of auditory stimuli was manipulated via changes to interaural intensity and timing (Whitworth & Jeffress, 1961; Wightman & Kistler, 1992)." The stimuli were delivered by a pair of loudspeakers, and the incident sound at each ear would be a product of both speakers. And - if there were a time delay between the two speakers, then both ears could potentially receive separate pulses one after the other at different delays. Did they record this audio stimulus with manikin? If not, it would be very difficult to know what it was at the ears. I don't doubt that if they altered the relative volume of the loudspeakers then some directionality would be perceived but I cannot see how the interaural level and timing differences could be matched - as if the sound were from a single source. I doubt that this invalidates their results, but to present this as if it provided matched spatial and timing cues is wrong, and I cannot work out how they can attribute an azimuthal location to the sound. For replication purposes, it would be useful to know how far apart the loudspeakers were and what the timing and level differences actually were.

The behavioural tasks each had evenly distributed ‘source locations’ on the horizontal azimuth of the computer display (8 for the behavioural session, 5 for the EEG session). We manipulated the perceived location of auditory stimuli through interaural time delays and interaural level differences. By first measuring the forward (*z*) and horizontal (*x*) distance of each source location to each ear, the method worked by calculating what the time-course of a sound wave should be at the location of the ear given the sound wave at the source. Then, for each source location, we can calculate the time delay between speakers given the vectors of *x* and *z*, the speed of sound and the width of the head. As the intensity of sound drops inversely with the square of the distance, we can divide the sound wave by the distance for each source location to provide the interaural level difference. Though we did not record the auditory stimulus with a manikin, our behavioural analyses show that participants were able to detect the directions of auditory stimuli from our manipulations, even to a degree that significantly exceeded the localisation accuracy for visual stimuli (for the behavioural session task). This information has been clarified in the manuscript.

“Auditory stimuli were played through two loudspeakers placed either side of the display (80 cm apart for the behavioural session, 58 cm apart for the EEG session).”

“The perceived source location of auditory stimuli was manipulated via changes to interaural level and timing (Whitworth & Jeffress, 1961; Wightman & Kistler, 1992). The precise timing of when each speaker delivered an auditory stimulus was calculated from the following formula:\begin{document}$$\displaystyle I T D_{L, R}=\frac{\sqrt{(x \pm r)^{2}+z^{2}}}{s}$$\end{document}

where *x* and *z* are the horizontal and forward distances in metres between the ears and the source of the sound on the display, respectively, *r* is the head radius, and *s* is the speed of sound. We used a constant approximate head radius of 8 cm for all participants. *r* was added to *x* for the left speaker and subtracted for the right speaker to produce the interaural time difference. For ±15° source locations, interaural timing difference was 1.7 ms. To simulate the decrease in sound intensity as a function of distance, we calculated interaural level differences for the left and right speakers by dividing the sounds by the left and right distance vectors. Finally, we resampled the sound using linear interpolation based on the calculations of the interaural level and timing differences. This process was used to calculate the soundwaves played by the left and right speakers for each of the possible stimulus locations on the display. The maximum interaural level difference between speakers was 0.14 *A* for ±15° auditory locations, and 0.07 *A* for ±7.5°.

I am confused about this statement: "A quantification of the behavioural sensitivity (i.e., steepness of the curves) revealed significantly greater sensitivity for the audiovisual stimuli than for the auditory stimuli alone (*Z* = -3.09, *p* = .002)," It is not clear from the methods how they attributed sound source angle to the sounds. Conceivably they know the angle of the loudspeakers, and this would provide an outer bound on the perceived location of the sound for extreme interaural level differences (although free field interaural timing cues can create a wider sound field).

Our analysis of behavioural sensitivity was dependent on the set ‘source locations’ that were used to calculate the position of auditory and audiovisual stimuli. In the behavioural task, participants judged the position of the target stimulus relative to a central stimulus. Thus, for each source location, we recorded how often participants correctly discriminated between presentations. The quoted analysis acknowledges that participants were more sensitive to audiovisual stimuli than auditory stimuli in the context of this task. A full explanation of how source location was implemented for auditory stimuli has been clarified in the manuscript.

It would be very nice to see some of the "channel" activity - to get a feel for the representation used by the decoder.

We have included responses for the five channels as a Supplemental Figure.

Figure 6 appears to show that there is some agreement between behaviour and neural responses - for the audiovisual case alone. The positive correlation of behavioural and decoding sensitivity appears to be driven by one outlier - who could not perform the audiovisual task (and indeed presumably any of them). Furthermore, if we were simply Bonferonni correct for the three comparisons, this would become non-significant. It is also puzzling why the unisensory behaviour and EEG do not correlate - which seems to again suggest a poor correspondence between them. Opposite to the claim made.

We understand the reviewer’s concern here. We would like to note, however, that each correlation used unique data sets – that is, the behavioural and neural data for each separate condition. In this case, we believe a Bonferroni correction for multiple comparisons is too conservative, as no data set was compared more than once. Neither the behavioural nor the neural data were normally distributed, and both contained outliers. Rather than reduce power through outlier rejection, we opted to test correlations using Spearman’s rho, which is resistant to outliers1. It is also worth noting that, without outlier rejection, the audiovisual correlation (*p* = .003) would survive a Bonferroni correction for 3 comparisons. The nonsignificant correlation in the auditory and visual conditions might be due to the weaker responses elicited by unisensory stimuli, with the reduced signal-to-noise ratio obscuring potential correlations. Audiovisual stimuli elicited more precise responses both behaviourally and neurally, increasing the power to detect a correlation.

(1) Wilcox, R.R. (2016), Comparing dependent robust correlations. *British Journal of Mathematical & Statistical Psychology, 69*(3), 215-224. https://doi.org/10.1111/bmsp.12069

“We also found a significant positive correlation between participants’ behavioural judgements in the EEG session and decoding sensitivity for audiovisual stimuli. This result suggests that participants who were better at identifying stimulus location also had more reliably distinct patterns of neural activity. The lack of neurobehavioural correlation in the unisensory conditions might suggest a poor correspondence between the different tasks, perhaps indicative of the differences between behavioural and neural measures explained previously. However, multisensory stimuli have consistently been found to elicit stronger neural responses than unisensory stimuli (Meredith & Stein, 1983; Puce et al., 2007; Senkowski et al., 2011; Vroomen & Stekelenburg, 2010), which has been associated with behavioural performance (Frens & Van Opstal, 1998; Wang et al., 2008). Thus, the weaker signalto-noise ratio in unisensory conditions may prevent correlations from being detected.”

Further changes:

(1) To improve clarity, we shifted the Methods section to after the Discussion. This change included updating the figure numbers to match the new order (Figure 1 becomes Figure 6, Figure 2 becomes Figure 1, and so on).

(2) We also resolved an error on Figure 2 (previously Figure 3). The final graph (Difference between AV and A + V) displayed incorrect values on the Y axis.

This has now been remedied.